# Flora Checklist in the Bayanaul State National Nature Park, Kazakhstan with Special Focus on New Species of Conservation Interest

**DOI:** 10.3390/plants14071119

**Published:** 2025-04-03

**Authors:** Zhumabekova Bibigul, Tarasovskaya Natalia, Klimenko Mikhail, Shakeneva Dinara, Assylbekova Gulmira, Shujaul Mulk Khan, Fazal Manan

**Affiliations:** 1Higher School of Natural Sciences, Margulan University, Pavlodar 140000, Kazakhstan; zhumabekovab@ppu.edu.kz (Z.B.); tarasovskaja_n_e@ppu.edu.kz (T.N.); klimenkomy@ppu.edu.kz (K.M.); shakeneva.dinara@mail.ru (S.D.); assylbekovag@mail.ru (A.G.); 2Department of Plant Sciences, Quaid-i-Azam University, Islamabad 45320, Pakistan; 3Pakistan Academy of Sciences, Islamabad 44000, Pakistan

**Keywords:** Bayanaul State National Nature Park, new record, endemic plants, systematics, medicinal plants, conservation efforts

## Abstract

Bayanaul State National Nature Park (BSNNP), which was established in 1985 and is one of the biggest natural parks in the Republic of Kazakhstan, conserves and rehabilitates the natural flora and fauna of the Bayanaul mountain range. This article expands the floristic inventory of BSNNP and identifies the ecological and ethnobotanical importance of the park. The literature revealed that 681 plant species inhabited the BSNNP region but it was hypothesized that the park’s plant diversity was greater than the documented 681 plant species. Following our expedition travels to BSNNP, we extended the flora summary with an addition of 81 new plant species. Now, according to this study, the flora of BSNNP comprises 762 plant species belonging to 335 genera and 81 families. The leading families are *Asteraceae* Dumort., *Poaceae* Barnhart, *Brassicaceae* Burnett, *Fabaceae* Lindl, *Rosaceae* Juss., *Caryophyllaceae* Juss, *Lamiaceae* Lindl., *Apiaceae* Lindl., and *Scrophulariaceae* Juss. They comprise 57.7% of the total plant species in the national park and 58.5% of the total genera. The largest genera are wormwood, sedge, onion, cinquefoil, speedwell, and astragalus, based on which these genera can be considered polymorphic. Moreover, 16 species of endemic plants belonging to 14 genera and 7 families were also reported. The flora is characterized by high biological diversity with the participation of boreal relicts. The largest group among useful species is medicinal plants, represented by 186 species (24.4%) belonging to 83 genera, and 39 families. Our findings enhance the scientific understanding of plant diversity in BSNNP and provide the groundwork for future conservation research.

## 1. Introduction

The floristic diversity of Kazakhstan, specifically within its protected areas, is a subject of increasing scientific interest due to its unique ecological characteristics and the urgent need for conservation efforts [1]. BSNNP, situated in the Pavlodar region, serves as an important habitat for various plant species, many of which are endemic and rare [1,2]. Several current studies at the global level, funded by municipalities, regions, ministries, and research institutions, have underscored the importance of documenting plant species to better understand biodiversity and inform conservation strategies, especially in some territories such as Kazakhstan, which are very rich in plant biodiversity. For example, Ref. [3] reported that Kazakhstan’s flora consists of 1406 species from 134 families, with significant representation from families such as Asteraceae and Rosaceae, which are also prevalent in the Bayanaul region [3]. This highlights the urgency of expanding the floristic inventory in regions like Bayanaul, where ecological assessments can reveal both the vulnerabilities and richness of local biodiversity.

The documentation of newly recorded plant species is necessary not only for academic purposes but also for practical conservation efforts [4]: the flora of Central Kazakhstan, including that of national parks, is important in understanding natural and anthropogenic alterations in these ecosystems [5]. The floristic studies conducted in different protected areas, such as the Buiratau State National Park, have employed systematic methodologies, including route reconnaissance and herbarium collections, to compile comprehensive species lists [6,7,8]. Such methodologies are crucial for ensuring that the floristic inventory of Bayanaul State National Nature Park is both accurate and reflective of the current ecological state. Moreover, the significance of biodiversity monitoring is emphasized in the context of human impacts on natural resources. References [9,10] highlight that biodiversity assessments are essential for identifying changes and threats to ecosystems, which is particularly relevant in the face of increasing anthropogenic pressures. In Kazakhstan, the need for updated floristic data is critical, as many regions, including Bayanaul, lack comprehensive studies that reflect current species compositions [11].

BSNNP flora is rich in medicinal plants, used for various disorders from ancient times. The parks have approximately 24.4% medicinal flora out of the total plant species [11]. These species are utilized by local inhabitants for different medicinal purposes, providing remedies for ailments ranging from gastric issues to respiratory infections. This utilization highlights the important role that plants play not only in ecological health but also in traditional heritage and livelihoods. Despite the benefits, the rising pressure on these plant species from tourism and habitat alteration poses a substantial risk to their conservation, requiring enhanced local legislative and administrative efforts to promote sustainable practices [11,12]. This knowledge gap can hinder effective conservation strategies, making the documentation of newly recorded species an urgent priority, also because these include many species useful to humans, such as food and medicine, on which botanists at a global level have started specific research for their valorization [13,14].

A critical issue of our era is the conservation of biological diversity, both in natural and artificial population-established reserves. Kazakhstan, as a contemporary nation, has ratified several UN conventions related to the preservation and judicious utilization of biodiversity since gaining independence, including the Rio Declaration on Environment and Development Declaration, (1992); the Convention on Biological Diversity; the Global Plan of Action for the Conservation and Utilization of Plant Genetic Resources for Food and Agriculture; and the International Treaty on Plant Genetic Resources for Food and Agriculture [15,16,17].

Being a unique natural site, Bayanaul State National Nature Park includes different ecosystems with high diversity, including rare relict and endemic species [6]. Some of these plants include *Glycyrrhiza uralensis*, *Betula pendula*, *Corydalis elegans*, *Artemisia gurguliana*, and *Eriophorum vaginatum*. Nowadays, it is a huge tourist area with a high number of visitors every year. The preservation of BSNNP is vital, and researchers are nowadays tasked with identifying methods to mitigate the extent of anthropogenic influence on this distinctive natural system [18]. Floristic research in Bayanaul commenced in 1816 with reports by I.P. Shangin’s expedition, published by G. Spassky, and has since progressed through excursions, organized scientific expeditions, project activities, dissertations, and independent studies. During this period, a substantial amount of information regarding the flora of Bayanaul National Park has been documented in the scientific literature, necessitating systematization and research. It was hypothesized that BSNNP’s plant diversity was greater than the documented 681 plant species. The objectives of this study were to (i) document and explore the newly recorded 81 plant species in Bayanaul State National Nature Park, aiding in an updated and inclusive floristic inventory of the region, (ii) identify and categorize the medicinal plants of newly recorded flora and document their traditional uses, and (iii) assess the ecological status of the newly identified species, highlighting the importance of conserving biodiversity in BSNNP.

## 2. Materials and Methods

### 2.1. Study Area

Bayanaul State National Nature Park is in the south-east of the Pavlodar region (Bayanaul district), 100 km from the industrialized city of Ekibastuz (Figure 1), on the eastern outskirts of the dry-steppe, Yereymentau-Bayanaul physical–geographical province, among dry steppes with dark chestnut soils of light mechanical composition. The province covers the vast northeastern part of Central Kazakhstan. It is characterized by a predominance of highly dissected hilly terrain with many blocky low mountains such as Bayanaul. The park encompasses a total area of 68,452.8 hectares.

BSNNP presents an exceptional environment characterized by significant climatic variability. The park’s climate is affected by its geographical location, with a mix of continental and steppe climates prevalent in the region. Previous studies show that the climate type in BSNNP is associated closely with a semi-arid climate, which is typical for much of Kazakhstan [19]. The park has long, cold winters and comparatively hot summers. Mean annual temperatures in Bayanaul usually range from approximately −12 °C in January to around 25 °C in July. Such temperature variations are characteristic of continental climates, which face stark seasonal variations [20]. The mean annual precipitation is around 300–400 mm, mostly rain during the late spring and summer months. This precipitation pattern aligns with the regional climate that highlights the increasing variability in rainfall along different areas of Kazakhstan, emphasizing the increased drought conditions attributed to climate change impacts in the larger region [20].

Remarkably, northern Kazakhstan, which comprises parts of BSNNP, has seen significant climatic variations, including a prominent decrease in precipitation (approximately 35% in recent decades) and an average increase in temperature by 1–2 °C, outstripping that of many other northern hemispheric regions. This climatic trend poses a challenge for maintaining biodiversity within BSNNP, as species that rely on certain climatic conditions are at risk due to habitat modification [20,21]. The integration of such data underscores the critical importance of monitoring climate variables in the park, as they naturally affect the flora and fauna diversity within this protected area [21].

### 2.2. Sampling

The materials and methods for the research was based on the flora of the Bayanaul State National Nature Park, published in the works of [22,23]. According to the previous studies, 681 plant species were reported in BSNNP. In addition, field research was carried out in 2023 on the territory of BSNNP using route reconnaissance and semi-stationary methods [6]). The semi-stationary method is used for continuous monitoring during certain conditions of natural territorial complexes, which are characteristic of it throughout the year. This method consists of studying several stationary points with different ecosystems or conditions at the same time. Regular visits are conducted to each data collection point, including changes and dynamics in environmental conditions, species abundance, and other observations throughout the year.

Initially, we categorized all natural regions in Bayanaul into multiple large clusters. These clusters are located near Bayanaul’s major lakes, i.e., Jasybay, Sabyndykol, Toraygyr, and Birzhankol. Furthermore, we selected various locations distant from lakes and popular hiking trails. A total of 69 sampling points were studied for floristic diversity. The size of each sampling plot was 1000 m^2^. The study locations were visited in two growing seasons, i.e., April to May and August to September, to record all the plant species of the area [24]. For this work, we created a protocol, which included different data on the area such as plant list, habitat, type of soil, coordinates, altitude, and cover range for various types of vegetation. At each point, we analyzed the floristic composition, considering the zonal variations of the landscape. We examined flora in the lowlands and on the slopes and summits of the mountains.

### 2.3. Plant Specimens’ Preservation and Identification

Plant species were recorded in the prepared lists in the locality, and their habitat type and other characteristics were also recorded. Those plant species that could not be identified at that time were collected with the help of a cutter, tagged, and placed in shopping bags. The plant specimens were then shade-dried and pressed with a plant pressor. Then, the plant specimens were poisoned using mercuric chloride plus ethyl alcohol solution [25]. Plant specimens were then identified by experts and were confirmed from the Flora of Central Kazakhstan [26]. The Latin names were clarified according to the summary by [27], and World Flora Online (https://www.worldfloraonline.org/ URL accessed on different occasions during the year 2024). We collected the literature data from modern sources; nevertheless, we used old classic works of past periods to trace the dynamics of Bayanaul’s flora. Until now, only 681 plant species had been reported in BSNNP [2,22,28,29].

### 2.4. Data Processing and Analyses

The plant list was prepared in MS Excel, and all related data were included. Plants’ names, families, and genera were confirmed from online websites, i.e., World Flora Online (https://www.worldfloraonline.org/), and the plant list was prepared. Furthermore, the ethnobotanical uses of plants were assessed and were classified as medicinal, decorative, feed, honey-bearing, soil, forest, phyto reclamation, food, technical, vitamin, essential oils, and insecticidal [30].

Ecological groups of plants were distinguished based on moisture content, which was calculated using a Scholander pressure chamber, one of the most widely used techniques for measuring plant water status. We fixed the pressure in branches and leaves and wrote them down in the protocol [31]. At the same time, we obtained data on climate control from the National Park Office, which included moisture, precipitation, and air pressure. All data were collected and analyzed during the fieldwork. Due to the research object, we determined the next ecological groups. Plant ectomorphs refer to living forms in connection to environmental variables; for terrestrial plants, this is predominantly humidity. Plants were categorized into xerophytes, mesophytes, and hygrophytes. A crucial aspect for plants is the degree of soil salt; vegetation thriving in saline soils was classified as halophyte ecomorph.

## 3. Results

### 3.1. Plant Species Diversity in BSNNP

The flora of BSNNP is represented by 762 species belonging to 335 genera and 81 families (Table 1 and Appendix A). Of the total plant species, 681 were previously reported from the park, while we reported 81 new species from BSNNP. *Equisetopsida* (Horsetails) were represented by 5 species, *Polypodiopsida* (Ferns) by 12 species, *Pinopsida* (Cone-bearing trees) by 2 species, Gnetopsida (Gnetovye) by 2 species, *Liliopsida* (Monocots) by 142 species, and *Magnoliopsida* (Dicotyledons) were represented by 599 plant species (Table 1).

### 3.2. Dominant Plant Families and Genera

The predominant families include *Asteraceae* Dumort. (124 species), *Poaceae* Barnhart (66 species), *Brassicaceae* Burnett (35 species), Fabaceae Lindl (49 species), *Rosaceae* Juss. (37 species), *Caryophyllaceae* Juss. (31 species), *Lamiaceae* Lindl. (29 species), *Apiaceae* Lindl. (26 species), *Scrophulariaceae* (26 species), and *Boraginaceae* Juss., (17 species (Figure 2 and Appendix A). These families constitute 57.7% of the total species and 58.5% of the overall genera in Bayanaul State National Nature Park.

The predominant genera include wormwood (25 species), sedge (18 species), onion (12 species), *astragalus* (14 species), cinquefoil (11 species), speedwell (11 species), and willow (10 species), indicating that the data genera can be classified as polymorphic. The genera *Poa*, *Calamagrostis*, and *Polygonum* each encompass 9 species (Figure 3 and Appendix A).

The vegetation of Bayanaul State National Nature Park has significant ecological diversity, featuring boreal relics such as *Neottianthe cucullata* L. Schlechter, *Ramischia secunda* (L.) Garcke, and *Dactylorhiza incarnate* L.

### 3.3. Habitat Types Based on Water Availability

All the plant species are classified as xerophytes, mesophytes, and hygrophytes. Plant ecomorphs refer to living forms in connection to environmental variables. For terrestrial plants, this is predominantly humidity. A crucial aspect for plants is the degree of soil salt. Vegetation thriving in saline soils is classified as halophyte ecomorph. The primary suite of ecomorphs comprises mesophytes, encompassing 303 species, representing 58.5%. The xerophyte suite comprises 187 species, accounting for 36.1% of the total. The assemblage of hygrophytes comprises merely 27 species (5.2%), with only 1 species classified as a typical halophyte (Figure 4).

### 3.4. Ethnobotanical Uses of Plant Species of BSNNP

The examination of flora within Bayanaul State National Nature Park revealed that 719 plant species were economically significant (Table 2). The predominant category of economically important species is the medicinal plants with 186 species (24.4%) belonging to 83 genera and 39 families. Ornamental plants occupy second place in species diversity, comprising 126 species belonging to 61 genera and 34 families. The highest diversity of fodder plant species is found within the *Poaceae*, *Fabaceae*, and *Asteraceae*. The most useful forage plants are species from the genera *Stipa*, *Poa*, *Festuca*, *Bromus*, *Glycyrrhiza*, *Astragalus*, *Trifolium*, *Medicago*, *Onobrychis*, and *Vicia*. These plants can function as forage for cattle and wildlife, in addition to being utilized for hay production.

### 3.5. Newly Recorded Plant Species and Their Habitat

Based on the results of our expedition trips to Bayanaul National Park, we supplemented the flora summary with the addition of 81 plant species. The newly recorded plant species, scientific names, families, vernacular names, and habitats are given in Table 3.

## 4. Discussion

Bayanaul State National Nature Park is classified as a gradual forest vegetation province, specifically within the dry steppe pine forest region of the Bayano-Karkaraly low mountains and the Bayanaul subdistrict of low mountain pine forests. The Bayanaul National Natural Park is situated in a compact mountain-steppe massif with distinct vertical zonation, predominantly including pine forests and a strongly continental climate. The soil inside BSNNP is correlated with vertical zoning influenced by the hilly topography. Mountain forest soil under pine woods and mountain black soils are present here. The topography of Bayanaul State National Park is markedly varied and extensively fragmented. The primary orographic features consist of low mountains and tiny hills characterized by sharp, irregular profiles with numerous rocky outcrops, interspersed with intermountain valleys. Relative elevations vary from 100 to 1027 m. Certain mountain ranges and rugged heights are classified as part of the middle mountains (Bayanaul Mountains—1027 m, Alabassky Mountains—700 m). The minor hills are in rows, with elevations ranging from 350 to 500 m. A dense network of narrow gorges and ravines characterizes the region’s landscape. The Bayanaul Mountains, by their origin, are classified within the erosion-tectonic relief group. The highest elevation attains 1027 m. The ecological dynamics of the park are due to the diverse topography, such as low mountains, hills, and intermountain valleys. The elevation ranges from 100 to 1027 m, with various microclimates and soil types, which promote the rich plant diversity observed. Due to the topography of the park, the composition of the soil is influenced by pine woods, for instance in the mountains, black soil, which ensures the availability of nutrients for the further support of various plant communities [32].

The results from our work on plant diversity in this area show a rich array of flora, with a total of 762 plant species identified, including 81 new species. This substantial addition to the present knowledge emphasizes the value of ongoing botanical surveys in comprehending and preserving biodiversity, given the need for comprehensive species recording in conservation efforts [33,34,35]. The predominance of certain families, especially Asteraceae (124 species) and Poaceae (66 species), represents wider ecological patterns observed in temperate regions, where these families are often well-signified owing to their adaptability and ecological flexibility. The ecosystem functions are mainly maintained by the diversity of species in these families by adopting the prevailing microclimatic variations [36]. Previous studies also identified that some of the given plant families were dominant in the study area. The dominant forest-forming species of the given families are Scots pine (*Pinus sylvestris*), which constitutes 75% of the entire wooded area. The residual forested area comprises primarily birch (*Betula pendula*) at 15.7%, aspen (*Populus tremula*) at 3%, and black alder (*Alnus glutinosa*) at 2.3%. Tree species in the park comprise 98% of the wooded area, while shrubs (such as *Spiraea crenata*, *S. hypericifolia*, *Rosa pisiformis*, *Juniperus sabina*, etc.) constitute 2%. The authors of [6] established the findings of the investigation into the species composition of Indigenous flora in the Bayanaul Mountains. The findings indicated that 16 species from 14 genera and seven families thrive in this region. Most of the species are perennial herbaceous plants and are classified as xerophytes according to their ecological associations. Six species of vascular plants were newly identified in Central Kazakhstan (*Centaurea pseudomaculosa* Dobrocz., *Cerinthe minor* L., *Equisetum × moorei* Newm., *Juniperus sibirica* Burgsd., *Lappula occultata* M. Pop., *Viola mirabilis* L.) and eight rare species in the region (Arctium *leiospermum* Juz. et Ye.V. Serg., *Equisetum scirpoides* Michx., *Geranium robertianum* L., *Linum corymbulosum* Rchb., *Poa remota* Forsell., *Ranunculus auricomus* L., *Silene dichotoma* Ehrh., *Veronica prostrata* L.) [2]. Another research project and analysis of the flora revealed that 542 species of vascular plants from 282 genera and 71 families inhabited the Ulytau Mountains (Central Kazakhstan). The predominant taxonomic families include *Asteraceae*, *Poaceae*, *Fabaceae*, *Rosaceae*, *Brassicaceae*, *Scrophulariaceae*, *Boraginaceae*, *Lamiaceae*, *Polygonaceae*, and *Apiaceae*. The ecological niche available within the park is determined by the diversity of genera, including wormwood (25 species) and sedge (18 species). Different environmental conditions influence these genera, advocating that the park’s habitats could provide a range of microenvironments favorable to plant growth [37,38,39].

The diversity of plants within BSNNP and their ethnobotanical uses play a key role in local economies and cultural practices. Among the recognized species, medicinal plants constitute the major category, with 186 species (24.4%) belonging to 83 genera and 39 families, highlighting the reliance of local communities on these plants for health and wellness [40]. Ornamental plants are the second most diverse group, with 126 species from 61 genera and 34 families, indicating the aesthetic and cultural values linked with local flora [41]. Additionally, the Poaceae, Fabaceae, and Asteraceae families contain the highest diversity of fodder plant species, specifically in the genera such as Stipa, Poa, and Glycyrrhiza. They provide important forage for livestock and wildlife and are a source of hay production [42]. In addition to enhancing the livelihoods of local communities, this diverse use of plant species determines how important it is to conserve plant biodiversity for cultural heritage and sustainable development [43]. The incorporation of traditional knowledge concerning these plants is critical for their conservation and balanced use, as it promotes an innate understanding of their ecological and economic importance [44,45,46]. Overall, the ethnobotanical richness of BSNNP demonstrates the relationship between humans and plants, reflecting the need for continued research work and conservation strategies to protect these precious resources and propagate further via various development strategies [47,48,49,50]. The outcomes of this study not only influence scientific understanding of plant diversity in BSNNP while also serving as a ground for future research directed at conserving this dynamic natural resource. Preserving the ecological integrity of the park and sustaining the lives of the local communities that depend on its biodiversity requires continued exploration and documentation of plant species.

## 5. Conclusions

This study concludes that studying plant diversity within Bayanaul State National Nature Park enhanced our understanding by identifying 762 species, including 81 new records of conservation interests. The findings confirm the ecological significance of the park’s varied topography, soil types, and climatic conditions in supporting a rich tapestry of flora, especially medicinal, ornamental, and fodder plants. The diversity and the ethnobotanical uses of BSNNP flora play a critical role in local livelihoods and cultural practices. Further research, sustainable use, and conservation efforts are essential to protect this biodiversity. Sustainable management of plant resources for upcoming generations while maintaining the park’s ecological integrity is also required. It is recommended to conduct a detailed disease-based ethnobotanical study on the plant species present in the park. Furthermore, a detailed conservation assessment of the endemic and rare plant species is necessary for the monitoring of proper conservation measures.

## Figures and Tables

**Figure 1 plants-14-01119-f001:**
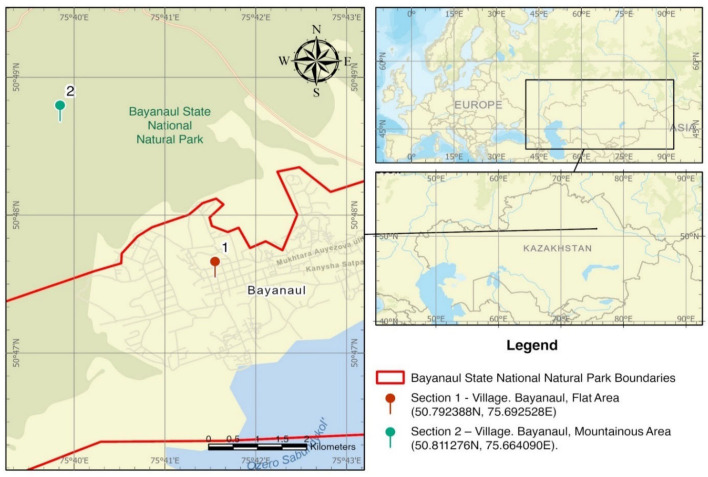
GIS-generated map of the Bayanaul State National Nature Park (the geo-referenced map is based on the WGS84 (World Geodetic System 1984) coordinate system).

**Figure 2 plants-14-01119-f002:**
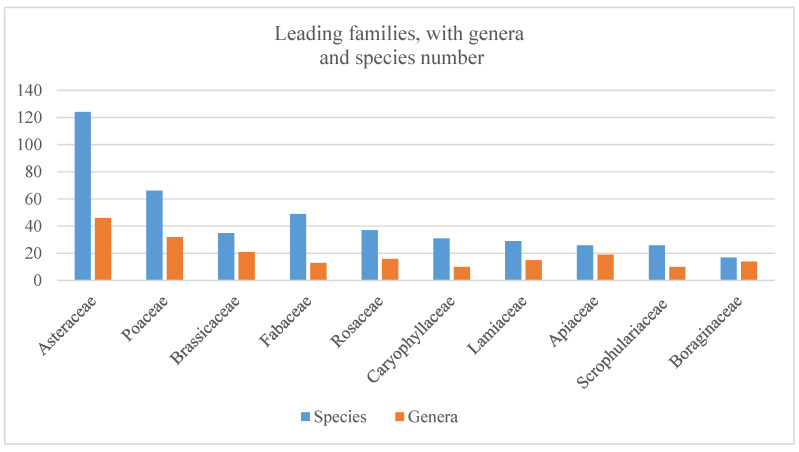
Leading plant families with the number of genera and species found at BSNNP.

**Figure 3 plants-14-01119-f003:**
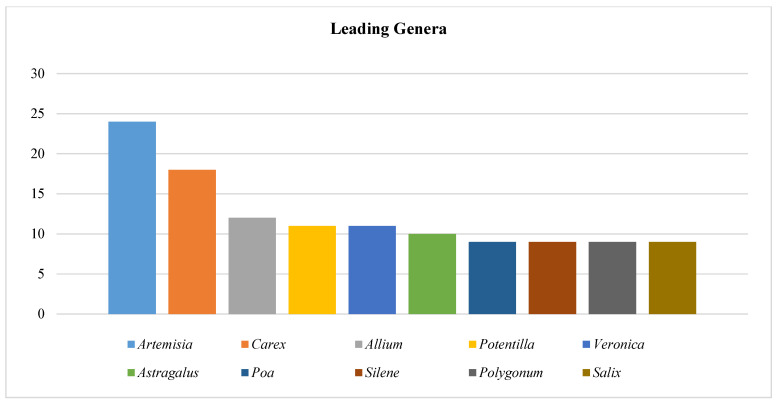
Spectrum of leading genera of plants in the Bayanaul State National Nature Park (number of species in the genus, pcs.).

**Figure 4 plants-14-01119-f004:**
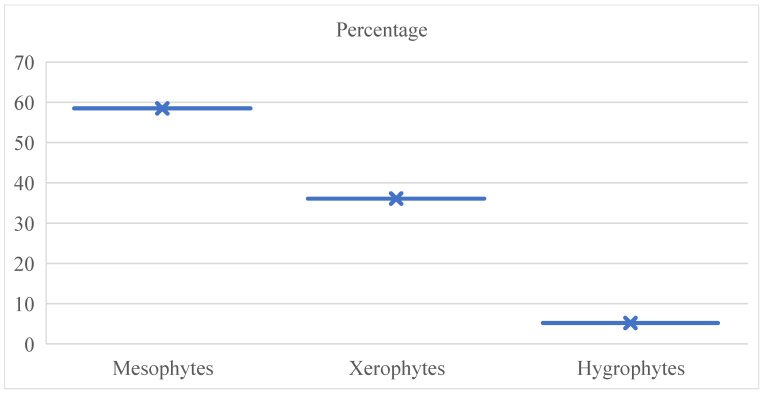
Bayanaul ecomorph types and their percentage.

**Table 1 plants-14-01119-t001:** Taxonomic characteristics of plants of Bayanaul State National Nature Park.

Taxon Name	No. of Families, pcs.	No. of Genera, pcs	No. of Species, pcs
Equisetopsida—Horsetails	1	1	5
Polypodiopsida—Ferns	7	10	12
Pinopsida—Cone-bearing trees	2	2	2
Gnetopsida—Gnetovye	1	1	2
Liliopsida—Monocots	14	54	142
Magnoliopsida—Dicotyledons	56	267	599

**Table 2 plants-14-01119-t002:** Groups of economically valuable plant species on the territory of BSNNP.

Economic Groups	Number of Species	%	Number of Births	%	Number of Families	%	Abundant Plant Species
Medicinal	186	24.4	83	24.8	39	48.1	*Glycyrrhiza uralensis*, *Betula pendula*, *Urtica dioica*, *Padus racemosa*, *Taraxacum officinale*, *Thymus serpyllum*, and *Solidago virgo-aurea*,
Decorative	126	16.5	61	18.2	34	42.0	*Gypsophila altissima*, *Gypsophila muralis*, *Iris pseudacorusm Veronica spicata*, *Pteridium aquilina*, *Lavathera turinginea*, and *Fritillaria* sp.
Feed	120	15.7	24	7.2	15	18.5	*Leymus akmolensis*, *Festuca sulcata*, *Agropyron pectinatum*, *Vicia sativa*, *Medicago falcata*, *Medicago sativa*, *Trifolium repens*, and *Bromus inermis.*
Honey-bearing	81	10.6	26	7.8	14	17.3	*Melilotus officinalis*, *Frangula alnus*, *Dracocephalus thymiflorum*, *Salvia spepposa*, *Eleagnus angustifolia*, *Trifolium pratense*, *Chamerion angusrifolia*, and *Caragana frutex.*
Soil, forest, phyto reclamation	60	7.9	37	11.0	22	27.2	*Pinus sylvestris*, *Populus balsamifera*, *Salix alba*, *Alnus glutinosa*, and *Glycyrrhiza uralensis.*
Food	54	7.1	11	3.3	9	11.1	*Grossularia* sp., *Ribes niger*, *Hippophae rhamnoides*, *Cirsium esculentum*, *Fragaria viridis*, *Rubus caesias*, and *Rubus idaeas.*
Technical	45	5.9	16	4.8	12	14.8	*Typha latifolia*, *Salix alba*, *Glycyrrhiza uralensis*, *Betula pendula*, *Pinus sylvestris*, *Lythrum salicaria*, and *Chamerion angusrifolia.*
Vitamin	22	2.9	5	1.5	4	4.9	*Rosa canina*, *Rosa cinnamomea*, *Sorbus sibirica*, and *Ribes niger.*
Essential oilseeds	21	2.8	14	4.2	5	6.2	*Thymus serpyllum*, *Ziziphora bungeana*, *Patrinia intermedia*, *Nepeta pannonica*, and *Nepeta ucranica.*
Insecticidal	4	0.5	4	1.2	3	3.7	*Pedicularis palustris*, *Anabasis aphylla*, *Tanacetum vulgare*, and *Artemisia absinthium.*

**Table 3 plants-14-01119-t003:** List of 81 new recorded plant species of high conservation interest at BSNNP.

No.	Plants	Family	Vernacular Name	Habitat and Distribution
1	*Pteridium aquilinum*	Dennstaedtiaceae	Common bracken	The most common background species of large ferns.
2	*Equisetum arvense*	Equisetaceae	Moore’s horsetail	Damp and open woodlands, pastures, roadsides, disturbed areas, and near the edge of streams.
3	*Ephedra intermedia*		Middle ephedra	Deserts, grasslands, floodlands and river valleys, slopes and cliffs, and sandy beaches.
4	*Aquilegia vulgaris*	Ephedraceae	Creeping buttercup	Wild species from the cultivated flora.
5	*Ranunculus repens*	Ranunculaceae	Creeping buttercup	Very common in damp places, ditches, and flooded areas.
6	*Thalictrum aquilegifolia*	Ranunculaceae	Basil foliage	Native range extends through Europe and temperate Asia, with a naturalized distribution in North America limited to New York and Ontario.
7	*Gypsophila perfoliata*	Caryophyllaceae	perfoliate baby’s-breath	Cultivated areas of gardens and parks.
8	*Stellaria holostea*	Caryophyllaceae	Chickweed	Widely distributed in broad-leaved and mixed forests.
9	*Atriplex nitens*	Chenopodiaceae	Quinoa	Adapted to dry environments with salty soils.
10	*Salicornia europaea*	Amaranthaceae	European saltwort	Grows in various zones of intertidal salt marshes, on beaches, and among mangroves.
11	*Camphorosma monspeliaca*	Amaranthaceae	Campharosma monspeliaca	Found in salt marshes and salty soils. Grows scattered or in groups.
12	*Halocnemum strobilaceum*	Amaranthaceae	Sarsazan knobby	A hyper-halophyte capable of growing alongside glasswort in conditions of extreme soil salinity.
13	*Kalidium foliatum*	Amaranthaceae	Potassium foliatum	Grow in saline mudflats, on alkaline soils, at margins of alluvial fans, and the shores of salt lakes.
14	*Anabasis aphylla*	Amaranthaceae	Leafless barnacle, itssigek	A many-branched shrub usually found growing in alluvial fans and dune swales.
15	*Limonium leptophyllum*	Plumbaginaceae	Elegant Kermek	The native range of this species is Central Asia. It grows primarily in the temperate biome.
16	*Populus nigra*	Salicaceae	Black poplar	Spreads mainly from plantings in populated and recreation areas.
17	*Populus alba*	Salicaceae	White poplar	Spreads from plantings in populated areas.
18	*Populus canescens*	Salicaceae	Gray poplar	Natural hybrid between aspen and white poplar, polymorphic and widespread, also used in decorative landscaping.
19	*Salix viminalis*	Salicaceae	Basket willow	It is a shrub or tree and grows primarily in the temperate biome.
20	*Ulmus pinnato-ramosa*	Ulmaceae	Pinnate elm	It is the last tree species encountered in the semi-desert regions of Central Asia.
21	*Urtica cannabina*	Urticaceae	Hemp nettle	Weed species in the northern regions spread from the south, mainly through a network of settlements and through accidental introduction.
22	*Euphorbia acuta*	Euphorbiaceae	Acute spurge	Found in arid and littoral conditions, they are typical xero- and thermophytes. Few species enter the tropical zone, and very few of them enter cold regions.
23	*Sedum purpurea*	Crassulaceae	Purple sedum	Typically found in dry habitats with sufficient bright sunlight.
24	*Sedum acre*	Crassulaceae	Caustic sedum	Typically found in dry habitats with sufficient bright sunlight.
25	*Ribes rubrum*	Crassulaceae	Red currant	Wild and wild cultivated forms.
26	*Grossularia reclinata*	Grossulariaceae	Rejected gooseberry, cultivated	Bred everywhere, runs wild, goes into foothill forests and grows together with native species.
27	*Syringa vulgaris*	Oleaceae	Common lilac	Cultivated around all populated areas.
28	*Fraxinus excelsior*	Oleaceae	Tall ash	Spreads from artificial plantings
29	*Pentaphylloides fruticosa*	Rosaceae	Kuril bush tea	It is a subshrub or shrub and grows primarily in the temperate biome.
30	*Pentaphylloides parviflora*	Rosaceae	Small-flowered Kuril tea	It is a subshrub or shrub and grows primarily in the temperate biome.
31	*Astragalus overiformis*	Fabaceae		Grow on the edges and clearings of larch, pine, birch forests, in mountain fir and spruce forests, steppes.
32	*Astragalus lanuginosus*	Fabaceae	Woolly astragalus	Grow on sands in light pine forests, as well as along riverbanks, on railway embankments, and along roadsides.
33	*Astragalus arenarius*	Fabaceae	Sandy astragalus	Grow on sands in light pine forests, as well as along riverbanks, on railway embankments, along roadsides.
34	*Astragalus glycyphillus*	Fabaceae	Astragalus licorice	Fresh, moist, slightly acidic, humus-rich, loamy and sandy loam soils.
35	*Glycyrrhiza glabra*	Fabaceae	Naked licorice	Grow in the valleys and floodplains of steppe and semi-desert rivers, on sand and shell ridges in the coastal zone, in the steppes and semi-deserts, in meadows.
36	*Hedysarym neglectum*	Fabaceae	Forgotten penny Weed	Grow in alpine and subalpine meadows, on grassy and rocky slopes, in juniper forests and on river pebbles.
37	*Lathyrus sylvestris*	Fabaceae	Forest chin	Found on forest edges and bush thickets.
38	*Lathyrus tuberosa*	Fabaceae	Chin nodule	Partly from the culture—perennial forms of sweet pea.
39	*Medicago sativa*	Fabaceae	Alfalfa	Grassy slopes, steppes, pastures, forest edges, bushes, gravel, river valleys, as a weed, in crops and around them.
40	*Caragana arborescens*	Fabaceae	Tree caragana	Partly spreads from artificial plantings
41	*Polygala sibirica*	Polygalaceae	Siberian source	Loose, well-permeable and water-permeable sandy soil. Heliophyte, drought-resistant.
42	*Erodium cicutarium*	Geraniaceae	Common stork	Found in fields, vegetable gardens, forest clearings, wastelands, and weedy places.
43	*Rhamnea frangula*	Geraniaceae	Brittle buckthorn	
44	*Eleagnus angustifolia*	Elaeagnaceae	Eleven angustifolia	It is undemanding to soils; it tolerates significant soil salinity and grows successfully on chestnut-solonetz, dark chestnut, and light chestnut soils.
45	*Eleagnus argentea*	Elaeagnaceae	Silver oleaster	It is undemanding to soils; it tolerates significant soil salinity and grows successfully on chestnut-solonetz, dark chestnut, and light chestnut soils.
46	*Hippophae rhamnoides*	Elaeagnaceae	Sea buckthorn	Sea buckthorn thickets are usually found in river floodplains and lake shores.
47	*Heracleum sosnowskii*	Umbelliferae	Sosnowski’s hogweed	Distributed everywhere along streams; unlike Siberian hogweed, it is not poisonous, does not have a burning effect, and serves as an edible plant.
48	*Pimpinella saxigraga*	Apiaceae	Saxifraga	Grow in meadows, in meadow steppes, among bushes, on forest edges, in sparse deciduous and pine forests, on hills.
49	*Laser trilobium*	Apiaceae		Lives in forests, forest edges, clearings, on limestone outcrops, on marls.
50	*Selinum carvifolia*	Umbelliferae	Gircha caraway	In meadows, on forest edges, under the canopy of damp broad-leaved, mixed, birch and coniferous trees, and in their clearings.
51	*Sium latifolium*	Umbelliferae		Along the banks of swamps, rivers, lakes, in peat pits and along the edges of reed-covered ditches.
52	*Sambucus racemosa*	Viburnaceae	Common elderberry	Spreads from artificial plantings.
53	*Symphoricarpos albus*	Caprifoliaceae	White snowberry	Spreads from artificial plantings in the vicinity of populated areas.
54	*Scabiosa scabiosoides*	Dipsacaceae	Common scabiosa	Across steppes and steppe meadows, pine forests, and mountain slopes.
55	*Valeriana officinalis*	Caprifoliaceae	Valerian	Along the banks of reservoirs, among thickets of bushes, in clearings and forest edges.
56	*Asperula odorata*	Rubiaceae	Woodruff	Grows in mixed and broadleaf forests, as well as in the forest-steppe zone and on riverine sands.
57	*Centaurlum umbellatum*	Gentianaceae	Centaury	Grow damp meadows, light forest edges, and between bushes.
58	*Convolvulus fruticosus*	Convolvulaceae	Shrub bindweed	Grows singly, like a weed, on successional slopes and wastelands.
59	*Symphytum officinale*	Boraginaceae	Comfrey	Spreads from artificial plantings, found in wastelands along with black root.
60	*Mertensia pallasi*	Boraginaceae		Endemic to Bayanaul, occasionally observed in floodplain and floodplain biotopes.
61	*Scrophularia nodosa*	Scrophulariaceae		Coniferous and mixed forests, among shrubs, on grass-forb, damp and dry meadows.
62	*Pedicularis palustris*	Orobanchaceae	Swamp grass	Wetlands, swamps, fens, marshes, wet meadows, and ditches.
63	*Plantago cornuti*	Plantaginaceae	Cornut’s plantain	Common inhabitant of salt marsh meadows.
64	*Leonurus cardiaca*	Lamiaceae	Motherwort	Becomes a weed flora from settlements where it was grown as a medicinal plant.
65	*Nepeta cataria*	Lamiaceae	Catnip	Roadsides, in vacant fields, waste ground, rubbish dumps, and other disturbed areas.
66	*Thymus serpyllum*	Lamiaceae	Creeping thyme	Sandy-soiled heaths, rocky outcrops, hills, banks, roadsides, and riverside sand banks.
67	*Tilia cordata*	Malvaceae	Heart-leaved linden	Common in the regional center and rural settlements, where it has taken root well in artificial plantings and is spread by birds.
68	*Artemisia procera*	Asteraceae	High wormwood	Inhabitant of dry meadows in steppe areas.
69	*Matricaria matricarioides*	Asteraceae	Chamomile without tongue	Blooming on footpaths, roadsides, and similar places in spring and early summer.
70	*Scorzonera orientalis*	Asteraceae	Eastern Kozelets	Grows on steppes, outcrops of various rocks, less often in pine forests, sometimes on sandy cliffs of riverside terraces.
71	*Potamogeton natans*	Potamogetonaceae	Floating pondweed	Native to quiet or slow-flowing freshwater habitats.
72	*Potamogeton crispus*	Potamogetonaceae	Curly pondweed	Native to quiet or slow-flowing freshwater habitats.
73	*Potamogeton lucens*	Potamogetonaceae	Shiny pondweed	Native to quiet or slow-flowing freshwater habitats.
74	*Potamogeton acutifolius*	Potamogetonaceae	Norway pondweed	Native to quiet or slow-flowing freshwater habitats.
75	*Scheichzeria palustris*	Scheuchzeriaceae	Swamp Scheichzeria	Grows on relict sphagnum bogs of sandy meadow terraces, in river valleys.
76	*Iris pseudacorus*	Iridaceae	Iris pseudacorus	Common in wetlands, where it tolerates submersion, low pH, and anoxic soils.
77	*Iris biglimus*	Iridaceae		Common in wetlands, where it tolerates submersion, low pH, and anoxic soils.
78	*Gagea lutea*	Liliaceae	Yellow goose onion	Broad-leaf woodlands, hedgerows, limestone pavements, pastures, and riverbanks.
79	*Asparagus setifora*	Asparagaceae	Bristle asparagus	Sources differ as to the plant’s native range but generally include most of Europe and western temperate Asia.
80	*Sparganum emersum*	Typhaceae	Pop-up hedgehog	Aquatic plant, growing in shallow water bodies.
81	*Leymus akmolensis*	Poaceae	Akmola hairweed	Found in saline meadows, salt marshes, gravel beds, and along roads.

## Data Availability

The original contributions presented in this study are included in the article/Appendix A. Further inquiries can be directed to the corresponding author(s).

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
