# Peer review of "Flora Checklist in the Bayanaul State National Nature Park, Kazakhstan with Special Focus on New Species of Conservation Interest"

_plants, 2025, doi:10.3390/plants14071119_

Round 1
Reviewer 1 Report
Comments and Suggestions for Authors
Dear Authors,
Your work was extensive and will certainly bring additional data on the flora of Kazakhstan. It is very good that in the species survey you also divided the species in an ethnobotanical sense and that you also wrote down their habitat types for newly recorded species.
In terms of methodology, it would be great if the research area were also defined from the point of view of climatic factors, the current description is lacking (add climate types, average precipitation, temperature...).
Also, add to the methodology how large were the individual survey localities (points) of the survey. Were they the same in terms of surface, different?
Also, in the usual survey of species in a certain area, the methodology is such that specimens are not collected but are determined on location, namely alive. Only if there is no other option, are they collected and herbarized.
Some other comments are marked in the manuscript.

Author Response
Comment 1: Your work was extensive and will certainly bring additional data on the flora of Kazakhstan. It is very good that in the species survey you also divided the species in an ethnobotanical sense and that you also wrote down their habitat types for newly recorded species.
Response 1: Thank you for your positive feedback and appreciation.
Comment 2: In terms of methodology, it would be great if the research area were also defined from the point of view of climatic factors, the current description is lacking (add climate types, average precipitation, temperature...).
Response 2: Thank you for pointing out this important portion. We have revised the study area introduction regarding climate type, precipitation, temperature, and climate change.
Comment 3: Also, add to the methodology how large were the individual survey localities (points) of the survey. Were they the same in terms of surface, different?
Response 3: A total of 69 sampling points were studied for floristic diversity based on habitat variability. The size of each sampling plot was 1000 m2.
Comment 4: Also, in the usual survey of species in a certain area, the methodology is such that specimens are not collected but are determined on location, namely alive. Only if there is no other option, are they collected and herbarized.
Response 4: The plants' whole data was recorded during the field visits, but those plants that were difficult to identify in the field were collected and brought for proper identification purposes.
Comment 5: Some other comments are marked in the manuscript.
Response 5: Thank you for your careful review. The given comments are addressed carefully in the manuscript, as suggested in the pdf file.

Reviewer 2 Report
Comments and Suggestions for Authors
The authors propose a manuscript titled “Floristic Inventory of the Bayanaul State National Natural Park, Kazakhstan; Documentation of 81 Newly Recorded Plant Species”. Particular attention was given on topic aspect on new wild plant species updating the floristic inventory in National Park of Kazakhstain that amount in 762 plant species especially of Asteraceae, Poaceae, Brassicaceae and Fabaceae families. A lot of number of these taxa are boreal relicts, or useful is medicine. I appreciate the original idea of the work which with a few revisions will convince me and the editor to publish it on Journal.
Title
Flora checklist in the Bayanaul State National Natural Park (Kazakhstan) with new species of conservation interest instead “Floristic Inventory of the Bayanaul State National Natural Park, Kazakhstan; Documentation of 81 Newly Recorded Plant Species”
- Introduction
Please add a references in the points that I have indicated because these are already known concepts, and in some cases complete (bold font) in the suggested way. Check whole introduction in this way.
- Lines 38-39. “BSNNP, situated in the Pavlodar region, serves as an 38 important habitat for various plant species, many of which are endemic and rare [choose reference]”;
- Lines 39-41. Several current studies at global level, funded by municipalities, regions, ministries, and research institutions, have underscored the importance of documenting plant species to better understand biodiversity and inform conservation strategies [choose reference], especially in some territory as Kazakhstan that are very rich in plant biodiversity [choose reference];
- Lines 47-48. The documentation of newly recorded plant species is necessary not only for academic purposes but also for practical conservation efforts [choose reference];
- Lines 61-62. In the abstract the authors declare the concept ogf medicinal wild plants but no information tehre is in introduction. Pleasecolpete in the suggested way. “This knowledge gap can hinder effective conservation strategies, making the documentation of newly recorded species an urgent priority, also because these include many species useful to human, such as food, medicine on which botanists at a global level have started specific research for their valorization [Ben Mahmoud et al. 2024; Perrino et al. 2024];
- Lines 71-72. Being a unique natural site, Bayanaul State National Natural Park includes different ecosystems with high diversity, including rare relict and endemic species [choose reference]. Please some examples of endemic or conservation interest species.
Reference to be added:
- Ben Mahmoud et al. 2024. Euro-Mediterranean Journal for Environmental Integration. https://doi.org/10.1007/s41207-024-00468-5
- Perrino et al. 2024. Planta. https://doi.org/10.1007/s00425-024-04571-3
- Materials and Methods
- Figure 1. Please complet in the text the geographical system used in the georeferenced map. WGS84? Gauss-Boaga? ….. Please specify
- Line 103. Please specify the sampling was realize in all seasons of the year 2023?
- The authors decalre that have studied 69 points with floristic diversity…. but with what criteria were chosen the 69 plot sampling? to cover all the vegetation environments in the area? Please explain better and how many times did you return to the same plot?
- Line 135. Please check font size for … https://www.worldfloraonline.org/
- Lines 147-148. Please explain the reason the choice to categorised the plants into xerophytes, mesophytes, and hygrophytes while abitually in the checklist or inventory plant the botanist used biological form as Therophyte, Chamaephyte ….
- Results and 4. Discussion
- Wel done, some suggetsions.
- Table 3.
Some species don’t report the family, as:
- Syringa vulgaris
- Fraxinus excelsior
- Erodium cicutarium
- Gagea lutea
- Some species are not reported in italic, as:
- Astragalus overiformis
- Conclusion
Please a few more words about the future prospects of research
Author Response
Reviewer 2
Comments and Suggestions for Authors
The authors propose a manuscript titled “Floristic Inventory of the Bayanaul State National Natural Park, Kazakhstan; Documentation of 81 Newly Recorded Plant Species”. Particular attention was given on topic aspect on new wild plant species updating the floristic inventory in National Park of Kazakhstain that amount in 762 plant species especially of Asteraceae, Poaceae, Brassicaceae and Fabaceae families. A lot of number of these taxa are boreal relicts, or useful is medicine. I appreciate the original idea of the work which with a few revisions will convince me and the editor to publish it on Journal.
Title
Flora checklist in the Bayanaul State National Natural Park (Kazakhstan) with new species of conservation interest instead “Floristic Inventory of the Bayanaul State National Natural Park, Kazakhstan; Documentation of 81 Newly Recorded Plant Species”
Response: Thank you for modifying the title. We have updated the title as suggested, and now the title is more attractive.
- Introduction
Please add a references in the points that I have indicated because these are already known concepts, and in some cases complete (bold font) in the suggested way. Check whole introduction in this way.
Response: The Reference and citation are added.
- Lines 38-39. “BSNNP, situated in the Pavlodar region, serves as an 38 important habitat for various plant species, many of which are endemic and rare [choose reference]”;
- Lines 39-41. Several current studies at global level, funded by municipalities, regions, ministries, and research institutions, have underscored the importance of documenting plant species to better understand biodiversity and inform conservation strategies [choose reference], especially in some territory as Kazakhstan that are very rich in plant biodiversity [choose reference].
- Lines 47-48. The documentation of newly recorded plant species is necessary not only for academic purposes but also for practical conservation efforts [choose reference];
- Lines 61-62. In the abstract the authors declare the concept ogf medicinal wild plants but no information tehre is in introduction. Please comlepete in the suggested way. “This knowledge gap can hinder effective conservation strategies, making the documentation of newly recorded species an urgent priority, also because these include many species useful to human, such as food, medicine on which botanists at a global level have started specific research for their valorization [Ben Mahmoud et al. 2024; Perrino et al. 2024];
- Lines 71-72. Being a unique natural site, Bayanaul State National Natural Park includes different ecosystems with high diversity, including rare relict and endemic species [choose reference]. Please some examples of endemic or conservation interest species.
Response: The above comments and suggestions are carefully addressed, And references are added.
Reference to be added:
- Ben Mahmoud et al. 2024. Euro-Mediterranean Journal for Environmental Integration. https://doi.org/10.1007/s41207-024-00468-5
- Perrino et al. 2024. Planta. https://doi.org/10.1007/s00425-024-04571-3
Response: Thank you for suggesting these relevant papers. They were helpful, and the references have been added to the manuscript.
- Materials and Methods
- Figure 1. Please complet in the text the geographical system used in the georeferenced map. WGS84? Gauss-Boaga? ….. Please specify
- Response: The georeferenced map is based on the WGS84 (World Geodetic System 1984) coordinate system.
- Line 103. Please specify the sampling was realize in all seasons of the year 2023?
Response: The study locations were visited in two seasons, i.e., April to May and August to September, to record all the plant species of the area following the methods of Kirby et al., (1986).
- The authors decalre that have studied 69 points with floristic diversity…. but with what criteria were chosen the 69 plot sampling? to cover all the vegetation environments in the area? Please explain better and how many times did you return to the same plot?
Response: The authors studied 69 points with floristic diversity, about 0.5 to 1 km apart from each other, depending upon the variability in the flora and habitat type. The study locations were visited in two growing seasons, i.e., April to May and August to September, to record all the plant species of the Park (Kirby et al., 1986).
- Line 135. Please check font size for … https://www.worldfloraonline.org/
- Response: Font size is corrected.
- Lines 147-148. Please explain the reason the choice to categorised the plants into xerophytes, mesophytes, and hygrophytes while abitually in the checklist or inventory plant the botanist used biological form as Therophyte, Chamaephyte ….
- Response: Our study was focused on habitats, that’s why we categorized the newly recorded plant species as xerophytes, mesophytes, and hygrophytes.
- Results and 4. Discussion
- Wel done, some suggetsions.
- Table 3.
Response: Thanks for your appreciation. Table 3 is modified.
Some species don’t report the family, as:
- Syringa vulgaris
- Fraxinus excelsior
- Erodium cicutarium
- Gagea lutea
- Some species are not reported in italic, as:
- Astragalus overiformis
Response: The families of these species have been added to the family column.
- Conclusion
Please a few more words about the future prospects of research.
Response: Suggested words/sentences have been added.

Reviewer 3 Report
Comments and Suggestions for Authors
This study is of interest and does not raise significant concerns. It has been conducted in the finest traditions of Soviet botany and contains a range of floristic analyses characteristic of botanical research from Soviet times. I do not consider this approach a shortcoming of the work; on the contrary, the scientific community currently needs studies of this kind. As in previous years, this article does not link the floristic list to the list of examined herbarium specimens. Such a linkage could be the subject of a separate study.
Apart from a few minor issues, I identify only one serious problem: This article is unsuitable for this journal. Similarly, submitting an article to the journal Nature would be a mistake. The MDPI Plants are intended to publish a different type of research. I advise the authors to find a more suitable journal.
I believe an excellent candidate for publication could be the Turczaninowia journal, which is published in Barnaul (Russia): http://turczaninowia.asu.ru/
Additionally, this paper could be successfully published in regional botanical magazines of Kazakhstan or Russia. However, in their response, the authors could enumerate specific scientific arguments that make this article publishable in Plants, which I might have overlooked. For example, it is not surprising that botanists overlooked 81 species in some particular areas of Kazakhstan. Why is this discovery of the broad interest?
Author Response
Author Response to the Reviewer 3 Comments:
Reviewer 3
Comments and Suggestions for Authors
This study is of interest and does not raise significant concerns. It has been conducted in the finest traditions of Soviet botany and contains a range of floristic analyses characteristic of botanical research from Soviet times. I do not consider this approach a shortcoming of the work; on the contrary, the scientific community currently needs studies of this kind. As in previous years, this article does not link the floristic list to the list of examined herbarium specimens. Such a linkage could be the subject of a separate study.
Response: Dear reviewers, thank you for your appreciation. Your comments and suggestions mean a lot to us.
Apart from a few minor issues, I identify only one serious problem: This article is unsuitable for this journal. Similarly, submitting an article to the journal Nature would be a mistake. The MDPI Plants are intended to publish a different type of research. I advise the authors to find a more suitable journal.
I believe an excellent candidate for publication could be the Turczaninowia journal, which is published in Barnaul (Russia): http://turczaninowia.asu.ru/
Additionally, this paper could be successfully published in regional botanical magazines of Kazakhstan or Russia. However, in their response, the authors could enumerate specific scientific arguments that make this article publishable in Plants, which I might have overlooked. For example, it is not surprising that botanists overlooked 81 species in some particular areas of Kazakhstan. Why is this discovery of the broad interest?
Response:
Dear Reviewer, I am thankful to you for your valuable feedback. We understand your concern about the suitability of our manuscript for the “Plants” journal. However, we would like to clarify why we believe our study aligns with the journal’s scope and contributes to the broader scientific community:
This paper is based on endemism, which is a topic of global interest. Endemism gives rise to floristic classification, floristic kingdoms, and floristic regions of the world, like Takhtajan classification of floristic region, etc. Therefore, the region is rich in endemic flora, and 81 species are new records to this region, and some of them are endemic. Therefore, it is a global concern as these plants are not present elsewhere in the world and may be of importance for scientists in terms of seed bank, gene bank, botanical garden, and herbarium developments. Therefore, we want this paper to be published in a journal of broader interest, and we think the MDPI “Plants” is a crucial journal for this.

Round 2
Reviewer 2 Report
Comments and Suggestions for Authors
Dear authors,
the last version of the manuscript has been improved correctly, but there is a last question about the bibliography before pubblication. The references must be reported with doi and complete name of all authors not only .... et al.
Regards,
Enrico
Comments on the Quality of English LanguageNo comment
Reviewer 3 Report
Comments and Suggestions for Authors
Refocusing the paper on conservation issues is a good idea. It makes the paper interesting for a broad audience and, therefore, publishable in Plants.